# A SNP-based capture and clustering workflow to assess donor-derived cell-free DNA in transplantation

Shigeki Mitsunaga[1,2], Yohei Yamada[1]*, Phuong Thanh Nguyen[3], Naoko Fujito[4], Hirofumi Nakaoka[5,6], Hiromichi Aoyama[7,8], Hiroshi Kitamura[9], Kenichi Saigo[8,10], Ituro Inoue[2], Akihiro Fujino[1], Masahiro Shinoda[11], Kazumasa Fukuda[12], Yuko Kitagawa[12]

1 Department of Pediatric Surgery, Keio University School of Medicine, Tokyo, Japan, 2 iSAN Bio Inc., Yokohama, Japan, 3 Institute of Biology, Vietnam Academy of Science and Technology, Hanoi, Vietnam, 4 Department of System Pathology for Neurological Disorders, Brain Research Institute, Niigata University, Niigata, Japan, 5 Department of Data Science, Kagoshima University Graduate School of Medical and Dental Sciences, Kagoshima, Japan, 6 Department of Cancer Genome Research, Sasaki Institute, Sasaki Foundation, Tokyo, Japan, 7 Department of Transplant Surgery, Japan Community Healthcare Organization Chiba Hospital, Chiba, Japan, 8 Toyo Medical Clinic Oami, Meiseikai Medical Corporation, Chiba, Japan, 9 Department of Pathology, National Hospital Organization Chiba-East Hospital, Chiba, Japan, 10 Department of Surgery, National Hospital Organization Chiba-East Hospital, Chiba, Japan, 11 Department of Hepato-Biliary-Pancreatic and Gastrointestinal Surgery, International University of Health and Welfare Narita Hospital, Chiba, Japan, 12 Department of Surgery, Keio University School of Medicine, Tokyo, Japan

* yohei.z7@keio.jp

## Abstract

Measurement of donor-derived cell-free DNA (dd-cfDNA) enables early, non-invasive monitoring of transplanted organs, including rejection detection. We developed a method to estimate dd-cfDNA ratios using capture hybridization of 300 SNPs, next-generation sequencing (NGS), and clustering analysis. Validation was conducted using simulated mixtures of fragmented genomic DNA from two individuals (0–100%). dd-cfDNA ratios were estimated via clustering, with and without 0% mixture samples to simulate the presence or absence of pre-transplant recipient plasma. When 0% samples were included, estimation achieved an $r^2$ of 0.9987 across the full 0–100% range; without them, $r^2$ remained high (0.9973) in the clinically relevant 0–10% range. The robustness of the method was further demonstrated by *in silico* downsampling. MAEs with 0% samples were 0.823%, 0.766%, and 0.702% at full, 50%, and 25% read depths, respectively (0–100% range). For the 0–10% range, MAEs were 0.333%, 0.300%, and 0.467% with 0% samples, and 0.413%, 0.367%, and 0.503% without them. These results indicate that the method maintains high accuracy even under reduced input and when pre-transplant data are unavailable. We also compared clustering-based estimates with direct calculations from kidney transplant recipients, where donor and recipient SNP genotypes were known. The concordance correlation coefficient (CCC) from day 0 to day 28 post-transplantation was 0.9887 and 0.9316 for unrelated pairs with and without pre-transplant data, respectively. For sibling pairs, CCCs were 0.9923 and 0.9675; for parent–child

**Data availability statement:** All relevant data are within the manuscript and its Supporting Information files. Sequencing data for the recipient cfDNA mimicking (spike-in) experiments are available from the NCBI BioProject repository under accession number PRJNA1219677.

**Funding:** This research was supported partly by AMED under Grant Number JP23ek0510040. The funders had no role in study design, data collection and analysis, decision to publish, or preparation of the manuscript.

**Competing interests:** Keio University, International University of Health and Welfare and iSAN Bio Inc. have filed a patent application on the subject matter of this publication: Patent application number 2025-089314. I.I. and S.M. are the co-founders of iSAN Bio Inc. None of the other authors have conflicts to declare.

pairs, the CCC was 0.9831 with pre-transplant data. CCC was not calculated for parent–child pairs without pre-transplant data due to limited samples (<10%, n = 3). These findings demonstrate high concordance, accuracy, and robustness of our clustering-based dd-cfDNA estimation method and support its potential utility in clinical transplantation settings.

## Introduction

Cell-free DNA (cfDNA) in plasma primarily consists of double-stranded DNA fragments approximately 150–180 base pairs in length, released during cellular apoptosis and necrosis [1,2]. Due to its origin from injured cells and its amenability to PCR amplification and sequencing, cfDNA has recently garnered attention as a non-invasive biomarker for monitoring acute rejection and graft injury in organ transplant recipients. This approach has been applied across various organ systems, including the kidney [3–5], liver [6–8], heart [9,10], lung [11,12], and pancreas [13,14].

Effective detection of graft injury requires distinguishing donor-derived cfDNA (dd-cfDNA) from recipient-derived cfDNA. Historically, this distinction has relied on genetic markers such as male-specific Y chromosome sequences [15,16] and highly polymorphic regions within the human leukocyte antigen (HLA) complex [16,17]. More recently, single-nucleotide polymorphisms (SNPs) and small insertions or deletions (indels) [18,19] have been increasingly utilized, including in commercial testing services.

Despite these advancements, tissue biopsy remains the gold standard for diagnosing rejection and assessing graft integrity [3]. Although protocol biopsies are routinely scheduled and episode biopsies are performed based on clinical or biomarker indications, cfDNA analysis offers the potential to reduce dependence on these invasive procedures, thereby minimizing patient burden [2]. This is particularly beneficial in organs such as the heart, where reliable circulating biomarkers remain limited [20,21].

Detection of dd-cfDNA typically involves PCR amplification of target SNPs. In multiplex PCR-based approaches, the primary limitation is primer–dimer formation and other interactions [22], whereas sequence-dependent amplification bias [23] is generally considered less problematic. Several computational algorithms have been developed to estimate dd-cfDNA fractions using target enrichment and next-generation sequencing (NGS). These methods often require mapping sequencing reads to a reference genome and may depend on pre-transplant genotyping information [24–26]. Such pipelines can be complex, reliant on empirical constants, and time-consuming due to the alignment process.

Here, we propose a novel method for estimating dd-cfDNA that bypasses genome mapping by directly quantifying read counts at predefined SNP loci [27,28] and applying a simple unsupervised clustering strategy [29,30]. This approach builds on a previously established capture hybridization protocol targeting 300 SNPs [5], enabling accurate quantification without PCR amplification and allowing for faster turnaround time.

## Materials and methods

### Ethical approval and sample collection

All procedures were conducted in accordance with the Declaration of Helsinki and were approved by the Ethics Review Committee of Keio University School of Medicine (Approval Number: 2023−1062). Written informed consent was obtained from all participants prior to the collection of their samples. Plasma samples were collected using Cell-Free DNA BCT CE tubes (Streck) and stored at −80°C until further processing. In some cases, the buffy coat fraction was also preserved for subsequent extraction of genomic DNA.

### SNP selection

We used a panel of 300 SNPs reported previously [5]. Briefly, SNPs were selected from all autosomes with minor allele frequencies (MAF) between 0.45 and 0.5, based on reported allele frequencies in the Japanese population [31]. Probes for capture hybridization targeting these SNPs were obtained from KAPA Biosystems as part of the KAPA HyperChoice Kit.

### cfDNA extraction and library preparation

Cell-free DNA (cfDNA) was extracted from 1 mL of plasma using the QIAamp MinElute ccfDNA Mini Kit (QIAGEN). The concentration of cfDNA was measured with a Qubit fluorometer (Thermo Fisher Scientific). DNA damage repair was performed using the NEBNext FFPE DNA Repair Mix, followed by library preparation with the NEBNext Ultra II DNA Library Prep Kit for Illumina (New England Biolabs). Libraries were amplified with 12 cycles of PCR and pooled prior to target enrichment. SNP capture was carried out using the HyperCap Target Enrichment Protocol (KAPA Biosystems), followed by 18 additional PCR cycles. Library quality was evaluated using either the Bioanalyzer system or the TapeStation system (Agilent Technologies). Sequencing was performed on the Illumina NovaSeq platform.

### Simulation of dd-cfDNA Using Genomic DNA Mixing

Genomic DNA was extracted from 200 µL of frozen buffy coat using the ReliaPrep Blood gDNA Miniprep System (Promega), and fragmented with NEBNext dsDNA Fragmentase to mimic the size distribution of cfDNA. Fragmented DNA exhibited smear patterns on TapeStation analysis, with peak fragment sizes ranging from 286 to 327 bp—comparable to the ~280 bp peak size typically observed for cfDNA. Fragmented genomic DNA from two individuals was mixed at varying ratios: 0%, 0.5%, 1%, 3%, 5%, 10%, 50%, and 100%. Library preparation and sequencing were performed as described above. The estimated mixing ratios obtained from the sequencing data were used to validate the dd-cfDNA quantification method. Validation data are available in the sheet titled "S1 Sheet 100% (full)" within the Supporting Information Excel file in S1 File.

To evaluate the robustness of the method with respect to variations in cfDNA input amounts, *in silico* downsampling was conducted by randomly reducing the number of reads to 50% and 25% of the original dataset using binomial sampling. We assessed the consistency of the estimated dd-cfDNA ratios using the mean absolute error (MAE), root mean square error (RMSE), and coefficient of determination ($r^2$). (see the sheet titled "S2 Sheet 50%" and "S3 Sheet 25%" in the Excel file provided as Supporting Information in S1 File).

**Note:** All R-squared ($r^2$) and concordance correlation coefficient (CCC) values are reported to four decimal places to preserve precision and reproducibility. These metrics typically yield values close to 1.000 in well-performing models, and rounding to fewer digits (e.g., three decimal places) may obscure meaningful variation and limit interpretability, especially when comparing across conditions or sample groups.

### Clinical validation using transplant genotype data

To assess the clinical utility of the method, we used previously published data [5] from kidney transplant recipients and their donors, for whom SNP genotypes were available. SNPs were selected based on homozygosity in the recipient for

either the reference or alternative allele. When the recipient was homozygous for the reference allele, the dd-cfDNA ratio after transplantation was considered equal to the increase in the alternative allele ratio. If the donor was heterozygous, the dd-cfDNA ratio was assumed to be twice this increase. Conversely, when the recipient was homozygous for the alternative allele, the dd-cfDNA ratio was defined as the decrease in the alternative allele ratio, or twice the decrease if the donor was heterozygous.

SNPs meeting these criteria were selected, and the dd-cfDNA ratios calculated for each SNP were averaged to determine the overall dd-cfDNA ratio for each sample. These directly calculated ratios were then compared with the dd-cfDNA estimates obtained through clustering analysis, both with and without the inclusion of pre-transplant reference data.

### Read count and SNP classification

The alternative allele ratio (ALT ratio) was calculated as follows:

ALT ratio = alternative allele read count/ (reference allele read count + alternative allele read count)

Read counts were obtained using the bbduk.sh command from the BBTools suite, with the following syntax:

bbduk.sh -da in=input.fastq ref=SNV_probes.fa rename=t k=31 mm=f grep -o @A clean.fq

This read counting process was accelerated by employing parallel processing via the parallel command.

SNPs were classified into three genotypes—AA, AB, and BB—based on the recipient's genotype, where "A" and "B" denote the reference and alternative alleles, respectively. Following transplantation, combinations such as AA/aa, AA/ab, and AA/bb were observed due to the introduction of donor-derived alleles.

### Estimation of dd-cfDNA Ratio with pre-transplant data

SNPs in the pre-transplant samples were classified according to their alternative allele ratio (ALT ratio) into three categories: 0.0–0.1 (AA), 0.4–0.6 (AB), and 0.9–1.0 (BB). The corresponding SNPs in the post-transplant samples were grouped using the same criteria. However, the heterozygous group (AB) exhibited substantial variability in ALT ratio and was therefore excluded from subsequent analyses.

The change in ALT ratio was calculated by subtracting the pre-transplant ratio from the corresponding post-transplant value. These differential values were then subjected to $k$-means++ clustering to form three groups (e.g., BB/bb, BB/ab, BB/aa) using $k = 3$. Clusters containing three or fewer elements were considered outliers and excluded. The remaining data were re-clustered, and dd-cfDNA ratios were calculated using the average ALT ratio within each cluster. Specifically, based on the average ALT ratios of each cluster, we calculated two values: (1) BB/bb – BB/aa and (BB/bb – BB/ab) × 2, and (2) AA/bb – AA/aa and (AA/ab – AA/aa) × 2. The average of these two values was then used as the estimated dd-cfDNA ratio for each sample. ("S1 Sheet 100% (full)" in the Supporting Information in S1 File). For parent–child transplants, where only partial allele mismatches are expected due to shared genetic background, $k$-means++ clustering was performed using $k = 2$.

### Estimation without pre-transplant data

In cases where pre-transplant data were not available, post-transplant SNPs were classified into three groups based on their alternative allele ratio (ALT ratio): 0.0–0.1 (AA), 0.4–0.6 (AB), and 0.9–1.0 (BB). As in the pre-data analysis, only the AA and BB groups were used for downstream analysis due to the variability observed in the AB group. Each of these groups was directly subjected to $k$-means++ clustering to generate three subgroups (e.g., BB/bb, BB/ab, BB/aa) with $k = 3$. Clusters containing three or fewer elements were considered outliers and excluded. Clustering was then repeated using the remaining data. The dd-cfDNA ratio was calculated using the mean alternative allele ratio of each resulting cluster, following the same approach as used in the presence of pre-transplant data. For parent-to-child transplants, $k$-means++ clustering was performed with $k = 2$, reflecting the expected allele sharing between donor and recipient. ("S4 Sheet 100%_wo_pre-data" in the Supporting Information in S1 File)

## Results

An overview of the methodology used in this study, including sample preparation, base sequence analysis, and dd-cfDNA estimation, is presented in Fig 1.

Representative alternative allele ratios from the "Simulation of dd-cfDNA Using Genomic DNA Mixing" (see Methods) are shown in Fig 2. Reference alleles are denoted as "A" and "a," and alternative alleles as "B" and "b," with uppercase letters indicating recipient-derived alleles and lowercase letters indicating donor-derived alleles. As shown in Fig 2A, alternative allele ratios in AB genotypes exhibited greater variability compared to homozygous genotypes (AA or BB). Therefore, only SNPs for which the recipient genotype was homozygous (AA or BB) were included in subsequent analyses. Figs 2B and 2C illustrate changes in alternative allele ratios for 1% and 3% mixtures, respectively, compared to the 0% mixture, which represents the pre-transplant condition. The 0% samples were categorized into three groups based on their alternative allele ratios: 0.9–1.0 (BB), 0.4–0.6 (AB), and 0.0–0.1 (AA). The same classification scheme was applied to the post-transplant (mixed) samples to ensure consistency in SNP selection.

The dd-cfDNA ratio was estimated by calculating the change in the alternative (or reference) allele ratio before and after transplantation. Specifically, for each SNP in each group, the allele ratio in the 0% mixture was subtracted from that in the corresponding mixed sample. The differences in alternative allele ratios were further divided into three subgroups using the $k$-means++ clustering algorithm. For recipients with the BB genotype, the resulting clusters corresponded to BB/bb, BB/ab, and BB/aa (Fig 3A); for those with the AA genotype, the clusters corresponded to AA/aa, AA/ab, and AA/bb (Fig 3B). The mixing ratio of fragmentase-treated genomic DNA, representing the dd-cfDNA fraction, was calculated based on the mean differential alternative allele ratio within each cluster. For example, in the 1% mixture, the average differential alternative allele ratios for BB/aa, BB/ab, and BB/bb were 0.0006, 0.0057, and 0.0134, respectively. Conversely, for AA/aa, AA/ab, and AA/bb, the corresponding values were −0.0010, −0.0081, and −0.0186. Similar trends were observed in the 3% mixture: 0.0009, 0.0129, and 0.0261

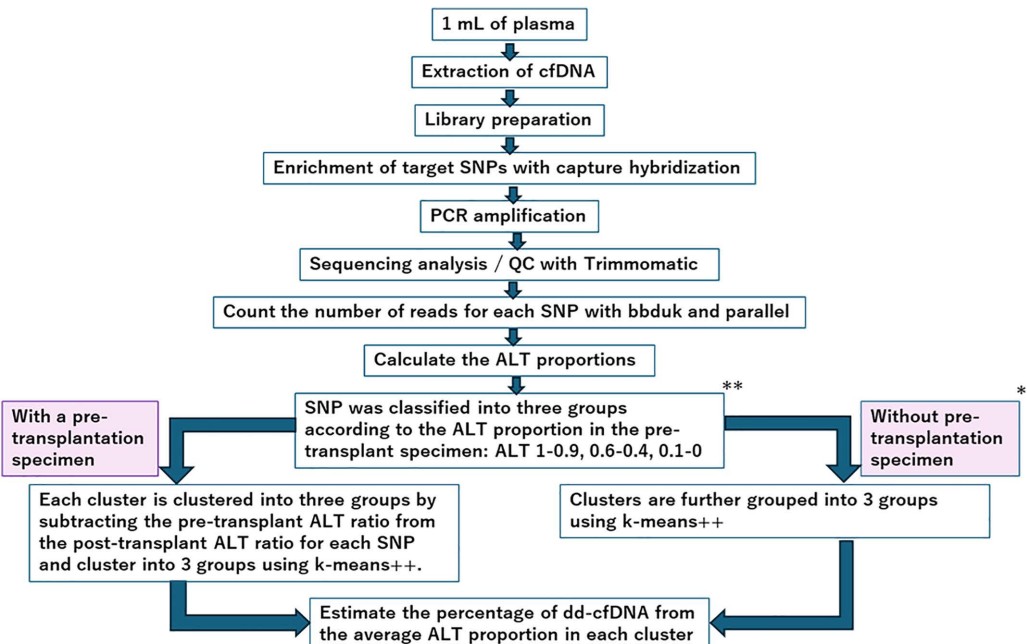

**Fig 1. Summary flowchart of the methods used in this study.** *If a pre-transplant recipient sample is not available, leukocyte-derived genomic DNA or DNA from other sources can be fragmentase-treated and used as a substitute pre-transplant sample. **Although SNPs are categorized into three groups based on ALT ratios, only two groups—ALT 0.9–1.0 and 0.0–0.1—are used in the subsequent analysis.

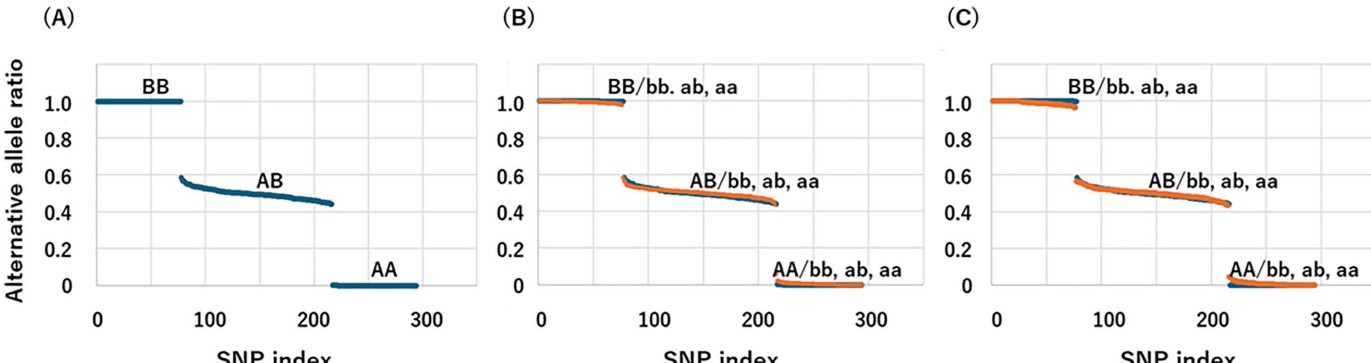

**Fig 2. Changes in alternative allele ratio in a mixing experiment using fragmentase-treated genomic DNA mimicking recipient cfDNA.** Fragmented genomic DNAs from two individuals were either left unmixed **(A)** or mixed at 1% **(B)** or 3% **(C)**. These samples underwent capture hybridization and NGS sequencing to determine alternative allele ratios. In each plot, SNPs are sorted in descending order of alternative allele ratio. In panels **(B)** and **(C)**, mixed samples are shown in orange and unmixed controls in dark blue. Reference alleles are denoted by A and a; alternative alleles by B and b. Uppercase letters indicate recipient-derived alleles, while lowercase letters indicate donor-derived alleles. The corresponding genotype combinations are illustrated in the figure.

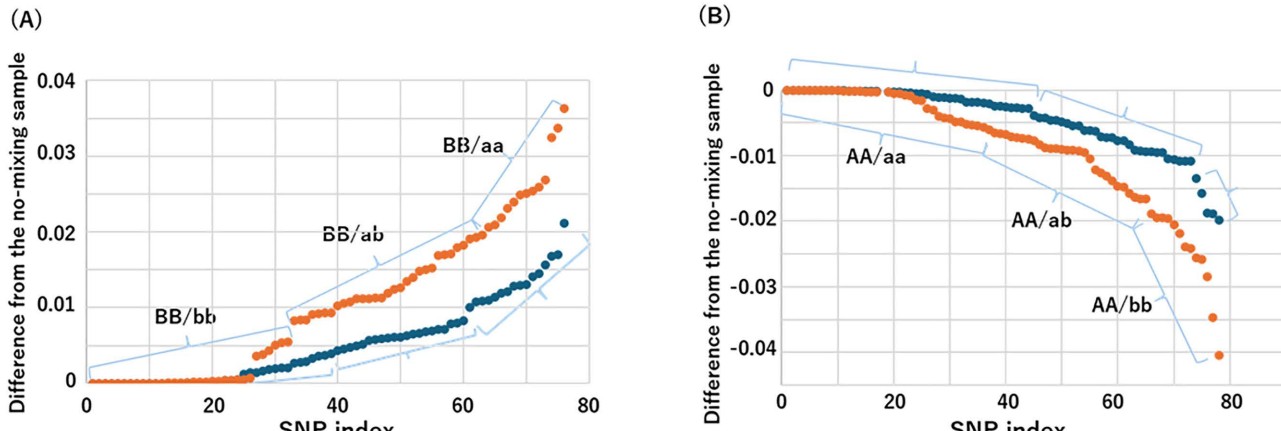

**Fig 3. Estimation of dd-cfDNA fraction using *k*-means++ clustering. (A)** Scatter plot of alternative allele ratio differences for all SNPs in which the recipient genotype is BB (n = 300). The data were grouped into three clusters using *k*-means++ clustering, corresponding to BB/bb, BB/ab, and BB/aa. **(B)** Scatter plot of alternative allele ratio differences for all SNPs in which the recipient genotype is AA (n = 300). The data were grouped into three clusters using *k*-means++ clustering, corresponding to AA/aa, AA/ab, and AA/bb.

for BB/aa, BB/ab, and BB/bb, and −0.0037, −0.0180, and −0.0386 for AA/aa, AA/ab, and AA/bb. These values support the biological consistency of the cluster assignments ("S1 Sheet 100% (full)" in the Supporting Information).

To evaluate the accuracy of dd-cfDNA estimation, we compared the estimated mixing ratios with the actual proportions of fragmentase-treated genomic DNA. Fig 4a displays the correlation between estimated and actual values across the full range of 0–100%. A strong linear relationship was observed, with a coefficient of determination ($r^2$) of 0.9987 and a regression equation of y = 0.9953x + 0.0059, indicating near-perfect concordance. ("S1 Sheet 100% (full)" in the Supporting Information in S1 File).

Focusing on clinically relevant dd-cfDNA levels below 10%, the correlation remained high. As shown in Fig 4B, the $r^2$ value was 0.9956 with a regression equation of y = 1.0006x + 0.0032, demonstrating that the estimation remains highly accurate within this lower range ("S1 Sheet 100% (full)" in the Supporting Information).

In clinical practice, pre-transplant samples are often unavailable, particularly when sample collection was not performed prior to transplantation. To address this limitation, we developed an alternative approach that does not rely on pre-transplant reference data. In this method, SNPs were categorized based on their observed alternative allele ratios into three groups: 0.9–1.0 (BB), 0.4–0.6 (AB), and 0.0–0.1 (AA), following the same classification strategy used with reference data. Cluster-based dd-cfDNA estimation was then performed using $k$-means++ clustering within each group.

As shown in Fig 4C, this reference-free method also yielded a strong correlation between estimated and actual values, with an $r^2$ of 0.9973 and a regression equation of $y = 0.8574x + 0.0032$. Although the slope was slightly reduced, indicating a modest underestimation, the high $r^2$ value supports the robustness of the approach even in the absence of pre-transplant data ("S4 Sheet 100%_wo_pre-data" in the Supporting Information).

The robustness of our method to variations in cfDNA input quantity is summarized in Table 1 and the Supplementary Information. Clustering-based dd-cfDNA estimates remained highly concordant, with $r^2$ values of 0.9988 (when using 0% mixing data to simulate pre-transplant samples) and 0.9964 (when not using 0% mixing data) for 50% downsampling ("S2 Sheet 50%" and "S5 Sheet 50%_wo_pre-data" in the Supporting Information), and 0.9994 and 0.9857, respectively, for 25% downsampling ("S3 Sheet 25%" and "S6 Sheet 25%_wo_pre-data" in the Supporting Information). These values were comparable to those obtained from full-read datasets (0.9987 and 0.9973, respectively). In terms of error, even with 50% and 25% downsampling, the MAE remained below 1% and the RMSE below 1.5% in the 0–100% range, and both metrics were below 0.5% in the clinically relevant 0–10% range (Table 1), supporting the robustness of the clustering-based estimation method under conditions of reduced cfDNA input.

Finally, we validated the clustering-based method using NGS data from transplant pairs with known donor and recipient SNP genotypes [5]. For each SNP where the recipient was homozygous for the reference allele and the donor was homozygous or heterozygous for the alternative allele (or vice versa), the average change in the alternative allele ratio after transplantation was used to calculate the dd-cfDNA fraction. In cases where the donor was heterozygous (e.g., parent-to-child transplants), the change in the alternative allele ratio was multiplied by two to reflect the contribution from both alleles.

These directly calculated dd-cfDNA values were compared with those estimated by our clustering-based method, which does not require donor genotype information (Fig 5, Table 2). Concordance between the two approaches was assessed

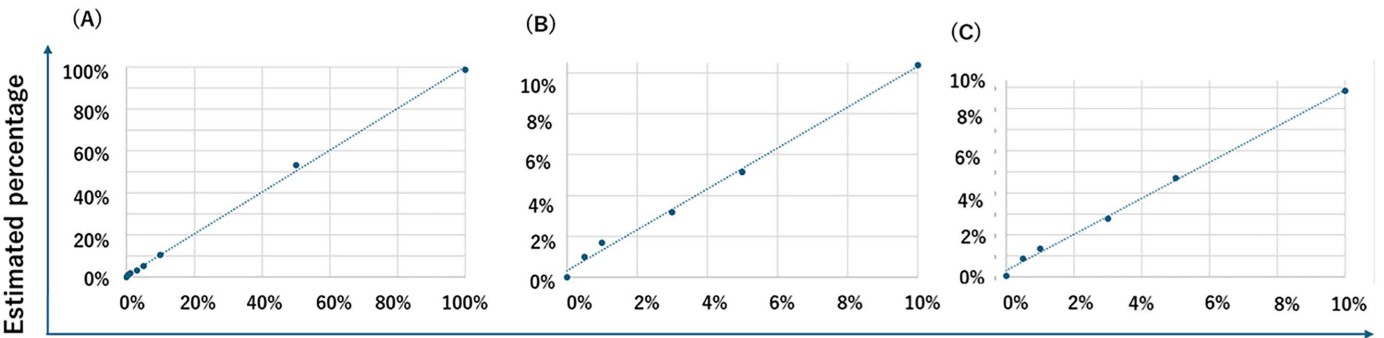

**Fig 4. Validation of dd-cfDNA estimation using genomic DNA mixing. (A)** Correlation between the actual mixing ratios of fragmentase-treated genomic DNA and the estimated dd-cfDNA fractions across the full range (0–100%). A strong linear relationship was observed ($r^2 = 0.9987$, $y = 0.9953x + 0.0059$). **(B)** Focused view of dd-cfDNA fractions within the clinically relevant range (0–10%), demonstrating continued high correlation ($r^2 = 0.9956$, $y = 1.0006x + 0.0032$). **(C)** Estimation of dd-cfDNA fractions without using pre-transplant reference data. SNPs were grouped based on observed alternative allele ratios, and clustering was applied within each group. The estimated values were highly correlated with the actual ratios ($r^2 = 0.9973$, $y = 0.8574x + 0.0032$), though with a slight underestimation ("S4 Sheet 100%_wo_pre-data" in the Supporting Information).

**Table 1. Assessment of Clustering-Based dd-cfDNA Estimation by *In Silico* Downsampling.**

| mixing rate | Estimated Mixing Rate Using 0% Sample | | | Estimated Mixing Rate Without 0% Sample | | |
|---|---|---|---|---|---|---|
| | Full* | 50%* | 25%* | Full* | 50%* | 25%* |
| 0.0% | 0.0% | 0.0% | 0.0% | 0.1% | 0.0% | 0.1% |
| 0.5% | 1.0% | 0.9% | 1.0% | 0.9% | 0.9% | 0.8% |
| 1.0% | 1.7% | 1.7% | 1.7% | 1.4% | 1.3% | 1.3% |
| 3.0% | 3.2% | 3.1% | 2.5% | 2.8% | 2.7% | 2.1% |
| 5.0% | 5.2% | 5.3% | 5.2% | 4.7% | 4.9% | 4.9% |
| 10.0% | 10.4% | 10.3% | 10.9% | 8.9% | 9.0% | 8.8% |
| 50.0% | 53.3% | 53.3% | 52.1% | | | |
| 100.0% | 98.7% | 99.0% | 99.3% | | | |
| $r^2$ | 0.9987 | 0.9988 | 0.9994 | 0.9973 | 0.9964 | 0.9857 |
| MAE (%)** | 0.823 | 0.766 | 0.702 | | | |
| MAE (%)*** | 0.333 | 0.300 | 0.467 | 0.413 | 0.367 | 0.503 |
| RMSE (%) | 1.308 | 1.266 | 0.906 | 0.539 | 0.477 | 0.643 |

*Downsampling was performed by randomly reducing sequencing reads to 50% or 25% of the original count.

**MAE for estimates in the 0–100% range.

***MAE for estimates in the 0–10% range.

Mixing rate represents the actual percentage of donor-derived DNA in the spike-in experiment. Estimated values were calculated using clustering methods applied to either the full dataset or downsampled data at 50% and 25% of original read depth. "Using 0%" indicates that the 0% mixing rate sample was included as a reference (e.g., pre-transplant baseline), while "without using 0%" excludes it.

using the concordance correlation coefficient (CCC). For unrelated donor–recipient pairs, CCC values from day 0 to day 28 post-transplantation were 0.9887 (with pre-transplant data) and 0.9316 (without pre-transplant data, excluding values ≥10%). For sibling pairs, CCC values were 0.9923 and 0.9675, respectively. For parent–child pairs, the CCC was 0.9831 with pre-transplant data; however, it could not be calculated without pre-transplant data due to the small number of data points below 10% (n = 3). These results demonstrate strong concordance across different transplant types and support the robustness of our clustering-based approach.

## Discussion

Building on the pioneering work of Snyder et al. [32], who first proposed donor-derived cfDNA as a non-invasive biomarker for transplant rejection, cfDNA-based monitoring of transplanted organs has now reached a level of practical application. In this study, we introduced a method for estimating dd-cfDNA by enriching 300 SNPs from plasma-derived cfDNA using capture hybridization, followed by clustering analysis of NGS data. To our knowledge, this is the first report to estimate dd-cfDNA using a clustering-based approach, which could, in principle, also be applied to PCR-amplified products.

To simulate post-transplant monitoring, we mixed fragmented genomic DNA from two individuals in varying proportions. The estimated dd-cfDNA ratios showed a strong correlation with actual mixing ratios ($r^2 = 0.9987$) across the full 0–100% range, demonstrating the high accuracy of the method when pre-transplant or surrogate genomic DNA is available. Even in the absence of pre-transplant data, the correlation remained robust ($r^2 = 0.9973$) within the clinically relevant range of 0–10%, indicating the method's practical utility. Although the accuracy decreased at higher dd-cfDNA levels (>10%) due to cluster overlap, this limitation is less concerning in clinical contexts because elevated dd-cfDNA levels are typically observed immediately after transplantation surgery, reflecting biological phenomena such as ischemia–reperfusion injury.

It should also be noted that small absolute deviations in the sub-1% dd-cfDNA range may result in proportionally large relative changes. For example, in the spike-in dataset, the 0.5% mixture was slightly overestimated at 1.0%. Such variation is partly attributable to sampling variance and Poisson noise at low allele counts and is not unique to this

## (A) Unrelated

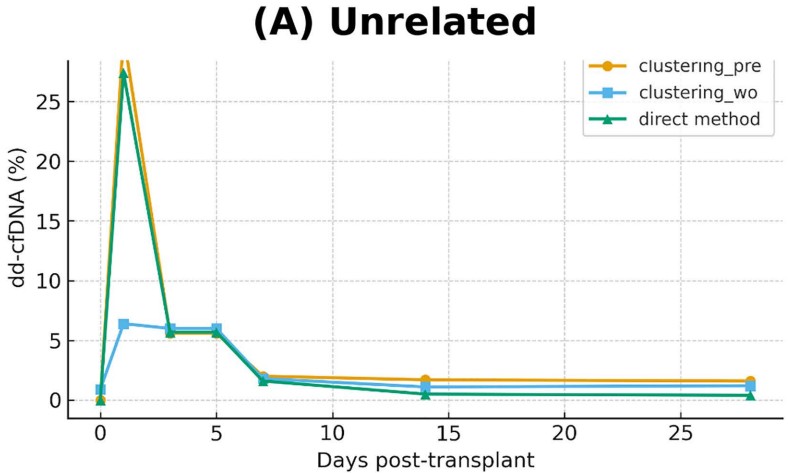

## (B) Sibling

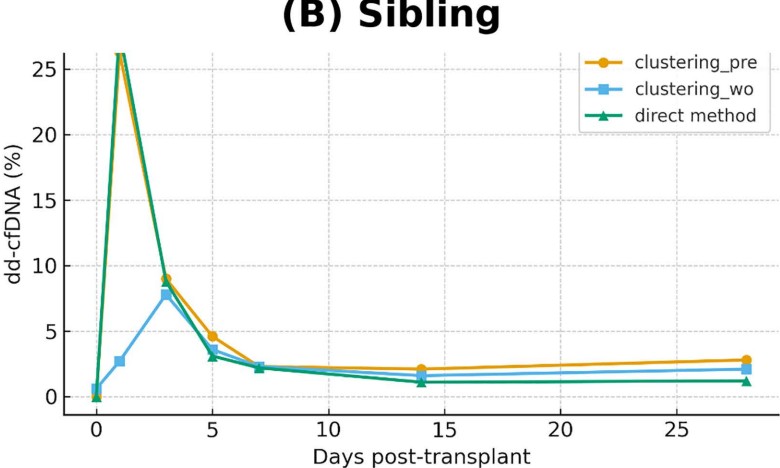

## (C) Parent–child

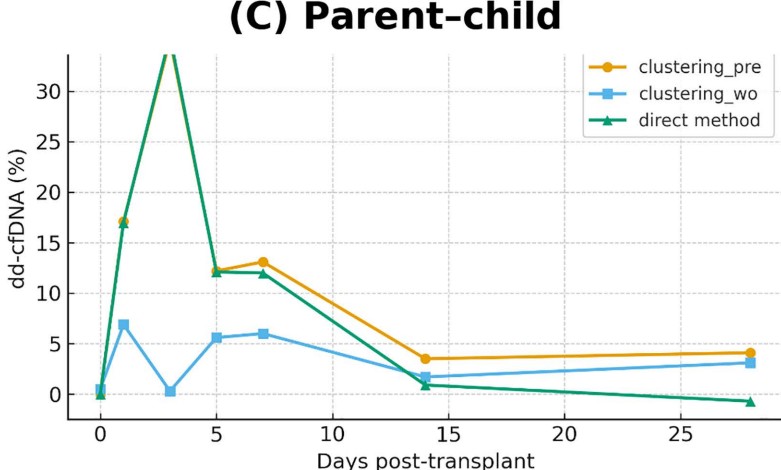

**Fig 5. Comparison of clustering-based estimation with direct genotype-based calculation.** Estimated dd-cfDNA ratios obtained using the clustering-based method with pre-transplant SNP data (orange:clustering_pre) and without pre-transplant SNP data (skyblue:clustering_wo) are compared to directly calculated values based on known donor and recipient genotypes (green:direct method). Each panel shows the time course of dd-cfDNA levels following kidney transplantation: **(A)** unrelated donor–recipient pairs, **(B)** sibling pairs, **(C)** parent–child pairs.

**Table 2. Comparison of dd-cfDNA ratios estimated using clustering-based methods and direct genotypic calculations.**

| Days after Tx | unrelated donor | | | sibling | | | parent to child | | |
|---|---|---|---|---|---|---|---|---|---|
| | direct method | clustering 1*1 | clustering 2*2 | direct method | clustering 1 | clustering 2 | direct method | clustering 1 | clustering2 |
| 0 | 0.0% | 0.0% | 0.9% | 0.0% | 0.0% | 0.6% | 0.0% | 0.0% | 0.5% |
| 1 | 27.4% | 30.6% | 6.4% | 27.9% | 26.3% | 2.7% | 17.0% | 17.1% | 6.9% |
| 3 | 5.7% | 5.6% | 6.0% | 8.8% | 9.0% | 7.8% | 35.5% | 35.1% | 0.3% |
| 5 | 1.5% | 2.7% | 2.5% | 3.1% | 4.6% | 3.6% | 12.1% | 12.2% | 5.6% |
| 7 | 1.6% | 2.0% | 1.8% | 2.2% | 2.3% | 2.3% | 12.0% | 13.1% | 6.0% |
| 14 | 0.5% | 1.7% | 1.1% | 1.1% | 2.1% | 1.6% | 0.9% | 3.5% | 1.7% |
| 28 | 0.4% | 1.6% | 1.2% | 1.2% | 2.8% | 2.1% | -0.7% | 4.1% | 3.1% |
| ccc*3 | 0.9887 | | 0.9316*4 | 0.9923 | | | 0.9675*4 | -*4 | |

*1 Clustering performed using pre-transplantation SNP data.

*2 Clustering performed without pre-transplantation SNP data.

*3 Concordance correlation coefficients (CCC) between clustering-based estimates and direct calculation.

*4 For clustering without pre-transplantation data, values ≥10% were excluded from CCC calculations due to reduced accuracy. In parent-to-child transplants, most values exceeded this threshold; therefore, CCC was not calculated.

clustering-based approach. Because some organ types, such as lung transplantation, use a threshold around 1% to flag potential rejection, values very close to this cutoff should be interpreted with caution and ideally evaluated in the context of longitudinal trends rather than single measurements.

One of the key challenges in clustering-based analysis is the handling of outliers. When pre-transplant samples were available, SNPs were initially classified by alternative allele ratios into three categories: 0.9–1.0 (BB), 0.4–0.6 (AB), and 0.0–0.1 (AA). SNPs with intermediate values (0.6–0.9 or 0.1–0.4) were excluded as outliers. In theory, and in the absence of sequencing noise, these three categories should correspond to homozygous alternative (1.0), heterozygous (0.5), and homozygous reference (0.0) genotypes. In practice, deviations from these ideal values are common; excluding outliers improved both clustering performance and concordance with clinical data.

For clustering based on differential alternative allele ratios, we applied the *k*-means++ algorithm, which allows for the explicit specification of the number of clusters. For instance, when the recipient genotype is homozygous BB, post-transplant cfDNA is expected to contain a mixture of BB/bb, BB/ab, and BB/aa, justifying the use of three clusters. In parent–child transplants, where only BB/bb and BB/ab combinations occur, clustering into two groups is more appropriate. Tests for normality revealed that alternative allele ratios within clusters did not follow a normal distribution, making the Smirnov–Grubbs test unsuitable for outlier detection. Although the interquartile range (IQR) method is a standard alternative, it proved ineffective in cases with small cluster sizes. In practice, clusters sometimes consisted of only one or two data points, which were therefore treated as outliers. As the IQR method failed to consistently detect these cases, we adopted a rule that excluded clusters containing three or fewer elements as outliers.

We validated our method using clinical samples from kidney transplant recipients with unrelated pairs, sibling, and parent–child donor relationships. High concordance was observed across all groups, as measured by the concordance correlation coefficient (CCC). The slightly lower CCC values observed in parent–child pairs likely reflect the exclusive presence of heterozygous SNP combinations (e.g., BB/ab, BB/bb, AA/aa, AA/ab), which exhibit greater variability than homozygous combinations. Indeed, in Fig 5c, the standard deviation of the alternative allele ratio for homozygous combinations ranged from 0.1% to 0.3%, compared to 3.7% to 5.0% for heterozygous ones. Nevertheless, the CCC of 0.9831 observed in parent–child transplants with pre-transplant SNP data remains well within a practically acceptable range.

It should be noted that our method does not rely on a trained predictive model. The clustering analysis was applied independently to each sample (or mixture ratio) without prior training or model fitting. No parameter optimization (e.g., filtering thresholds or cluster centroids) was performed using the spike-in dataset. As this is an unsupervised clustering method applied separately to each dataset, there was no need for data splitting or cross-validation typically required in supervised learning. Therefore, the high $r^2$ observed in the spike-in dataset reflects the inherent separability of allele frequency distributions rather than model overfitting.

This study was conducted as part of Transplant Medical Technology Development Project supported by Japan Agency for Medical Research and Development (AMED), specifically within an initiative to develop diagnostic methods for graft injury using donor-derived cell-free DNA (liquid biopsy) based on Japanese SNPs. To date, over 650 samples across five organ types have been analyzed using this method in both longitudinal and cross-sectional cohorts. While detailed clinical outcomes are being reported separately by participating institutions and are therefore not shown here, overall trends—such as elevated dd-cfDNA levels shortly after transplantation due to ischemia–reperfusion injury [33], and stable levels during non-rejection periods—were consistent with previously published findings. Moreover, dd-cfDNA level fluctuations observed in relation to rejection status showed good concordance with biopsy-based assessments [34–38]. These observations provide indirect support for the clinical relevance of this clustering-based estimation method.

Several commercially available dd-cfDNA assays, such as AlloSure and Prospera, have demonstrated clinical utility and are widely used without requiring donor genotyping. These platforms typically rely on proprietary algorithms and genome-wide or large-panel sequencing approaches, enabling robust performance across diverse clinical settings.

In contrast, the present study focuses on a targeted SNP capture strategy combined with a simple clustering-based analytical framework. Rather than aiming to directly replace existing commercial assays, our approach is intended as a complementary and transparent workflow that can be implemented on standard benchtop NGS platforms using open-source software.

Although a direct head-to-head comparison of analytical performance, turnaround time, and cost with commercial platforms was beyond the scope of this study, such comparative evaluations represent an important next step. Future prospective studies evaluating a subset of samples using both our method and established commercial assays will be essential to clarify relative strengths, limitations, analytical concordance, and appropriate clinical use cases.

In addition, while a formal cost analysis was not performed, the economic feasibility of this workflow warrants consideration. By relying solely on targeted SNP capture and straightforward clustering of allele frequencies, the proposed method eliminates donor genotyping and genome-wide mapping steps, thereby reducing reagent consumption and analytical overhead. Because the workflow can be implemented on standard benchtop NGS platforms without proprietary software or specialized computational infrastructure, it may offer advantages in operational simplicity and scalability. However, direct quantitative comparisons of cost and efficiency will require dedicated future studies.

An important limitation of the present approach is the potential measurement uncertainty in the low dd-cfDNA range, particularly below approximately 1%. At very low allele fractions, stochastic variation in read sampling and cluster assignment can lead to increased relative error, as illustrated by the experimental mixture in which a nominal 0.5% dd-cfDNA fraction resulted in an estimated value of approximately 1.0%. Such variability may be clinically relevant in transplant settings where decision thresholds around this range are commonly used.

Accordingly, dd-cfDNA values near the lower detection limit should not be interpreted in isolation but rather in the context of longitudinal trends and overall clinical findings. In this regard, the primary strength of the clustering-based framework lies in its ability to capture relative changes over time within the same patient, rather than providing absolute precision at the lowest detectable range. Future methodological refinements, increased sequencing depth, and prospective validation studies will be required to further improve accuracy and robustness in this clinically sensitive low-level range.

## Supporting information

**S1 File. Estimating mixing ratio of dd-cfDNA.** This Excel file contains six sheets detailing the calculation of dd-cfDNA ratios from raw read count data. Sheet 1: 100% (Full). dd-cfDNA ratios calculated using undownsampled read count data, with the 0% mixture representing the pre-transplantation state. Results are shown in Figures 4A, 4B, and Table 1. Sheet 2: 50%. dd-cfDNA ratios calculated using read count data downsampled to 50%, with the 0% mixture representing the pre-transplantation state. Results are shown in Table 1.
(XLSX)

**S1 Table. List of 300 SNPs used in this study.**
(XLSX)

## Acknowledgments

This research was supported partly by AMED under Grant Number JP23ek0510040. We thank J. Kitayama (National Institute of Genetics), and M. Sugimoto (Keio University School of Medicine) for excellent technical assistance and helpful discussion.

## Author contributions

**Conceptualization:** Shigeki Mitsunaga, Phuong Thanh Nguyen, Naoko Fujito.

**Data curation:** Kenichi Saigo.

**Funding acquisition:** Yohei Yamada, Ituro Inoue, Yuko Kitagawa.

**Investigation:** Shigeki Mitsunaga, Yohei Yamada, Phuong Thanh Nguyen, Naoko Fujito, Hirofumi Nakaoka, Hiroshi Kitamura, Kenichi Saigo, Ituro Inoue, Kazumasa Fukuda.

**Methodology:** Hirofumi Nakaoka, Kazumasa Fukuda.

**Project administration:** Yohei Yamada, Masahiro Shinoda, Yuko Kitagawa.

**Resources:** Hiromichi Aoyama, Hiroshi Kitamura, Kenichi Saigo, Akihiro Fujino.

**Software:** Shigeki Mitsunaga, Naoko Fujito, Hirofumi Nakaoka.

**Supervision:** Kenichi Saigo, Ituro Inoue, Akihiro Fujino, Masahiro Shinoda, Kazumasa Fukuda, Yuko Kitagawa.

**Writing – original draft:** Shigeki Mitsunaga, Phuong Thanh Nguyen.

**Writing – review & editing:** Yohei Yamada, Phuong Thanh Nguyen, Naoko Fujito, Hirofumi Nakaoka, Hiromichi Aoyama, Hiroshi Kitamura, Kenichi Saigo, Ituro Inoue, Akihiro Fujino, Masahiro Shinoda, Kazumasa Fukuda, Yuko Kitagawa.

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
