## [Decision Letter · Decision Letter 0]

3 Oct 2025

Dear Dr. Yamada,

Thank you for submitting your manuscript to PLOS ONE. After careful consideration, we feel that it has merit but does not fully meet PLOS ONE’s publication criteria as it currently stands. Therefore, we invite you to submit a revised version of the manuscript that addresses the points raised during the review process.

We look forward to receiving your revised manuscript.

Kind regards,

Elingarami Sauli, PhD

Academic Editor

PLOS ONE

Journal Requirements:

3. Please expand the acronym “AMED” (as indicated in your financial disclosure) so that it states the name of your funders in full.

This research was supported partly by AMED under Grant Number JP23ek0510040.

This research was supported partly by AMED under Grant Number JP23ek0510040. We thank J. Kitayama (National Institute of Genetics), and M. Sugimoto (Keio University School of Medicine) for excellent technical assistance and helpful discussion.

This research was supported partly by AMED under Grant Number JP23ek0510040.

Reviewers' comments:

Reviewer's Responses to Questions

**Comments to the Author**

1. Is the manuscript technically sound, and do the data support the conclusions?

Reviewer #1: Yes

Reviewer #2: Yes

2. Has the statistical analysis been performed appropriately and rigorously?

Reviewer #1: Yes

Reviewer #2: I Don't Know

3. Have the authors made all data underlying the findings in their manuscript fully available?

Reviewer #1: Yes

Reviewer #2: Yes

4. Is the manuscript presented in an intelligible fashion and written in standard English?

Reviewer #1: Yes

Reviewer #2: Yes

Reviewer #1: Dr. Mitsunaga and colleagues present an excellent paper introducing a novel method for test-ing donor-derived cell-free DNA in transplantation patients.

The study provides a precise measurement method for dd-cfDNA quantification using a clus-tering system for NGS-based SNP data. A highly interesting approach. And an approach which can be used with or without pre-transplantation data.

This is a very interesting field, and the methodology and testing precision are key elements for the prospects of its clinical application. Therefore, this study is highly relevant.

I find the study extremely well formulated and presented, and I have only minor comments for the authors to consider.

Minor considerations

1. Abstract (page 3). I find that some of the content in the abstract is a little bit difficult to un-derstand, when not having read the full paper yet. Consider trying to make it more readable, although this is likely not easy.

2. Introduction (page 6, lines 88-91). Here, you provide some information about different techniques. This is perhaps a little too short. There are other strengths and weaknesses which could be mentioned beside multiplexing. Consider if you should elaborate a little on this topic.

3. Materials and Methods (page 7, lines 108-110). This is just a detail. Please specify that blood (and not plasma) is collected in Streck tubes, and that (only) plasma is stored (presum-ably in another type of tube) at -80 Degrees C.

4. Materials and Methods (page 8, lines 136-137). Here, you present the fragment sizes after treatment with fragmentase. It might be standard, but I find that the fragments are perhaps a little bit large? In any case, it would be great to include a reference for your statement on these fragments lengths being comparable to what is typically observed.

5. Materials and Methods (page 9, lines 151-156). I completely agree with the content of your note, but I think it would be ok to simply delete the note.

6. Table 2 (page 23). So, these data in Table 2 show excellent correlation as you demon-strate. But I think the data differences on day 1 are substantial between clustering 1 and 2. Consider if you should add a discussion on these differences (in the discussion).

7. Discussion / For comparison with other methods, it could be very interesting to hear about the turnaround time of the method (what is the time duration from the top to bottom of figure 1?).

Reviewer #2: 1. Clinical Correlation and Utility

The article lacks direct correlation analysis with clinical outcomes (e.g., biopsy-proven rejection, renal impairment). Given that data from over 650 samples are available, it is recommended that the authors include a preliminary analysis—such as a graph showing the dynamic changes in dd-cfDNA (as detected by this method) in one or two cases during rejection events. This would significantly enhance the article’s persuasiveness and clinical relevance.

Additionally, while Table 2 demonstrates concordance between clustering-based estimates and direct genotypic calculations, the manuscript does not compare the sensitivity and specificity of the clustering method versus traditional methods in diagnosing rejection episodes. Including such a comparison would better highlight the clinical utility of the approach.

2. Cost and Efficiency

The introduction mentions that existing NGS-based methods for estimating dd-cfDNA ratios are time-consuming. While the proposed clustering method appears to reduce analysis time, the manuscript does not discuss cost advantages over traditional approaches. Could the authors provide a comparative analysis of the costs involved in their method versus conventional techniques?

3. Population Specificity and Generalizability

The 300 SNPs selected were based on their frequencies in the Japanese population. It is unclear whether these polymorphisms are equally effective in other ethnic groups (e.g., Caucasians, Africans). The generalizability of the method and any adjustments that may be needed for its application in different populations should be discussed.

4. SNP Panel Design and Sequencing Parameters

As discussed in recent literature[1], several factors critically impact dd-cfDNA assay performance:

Nucleosome Footprinting: The authors should clarify whether their 300-SNP panel avoids regions with strong nucleosome positioning, as differential nucleosome occupancy between donor and recipient tissues can lead to significant quantification bias.

SNP Count and Sequencing Depth: The choice of 300 SNPs and the achieved sequencing depth are not thoroughly justified. The authors should reference theoretical calculations to demonstrate that their panel size and depth are sufficient to reliably detect low-frequency heterologous signals (e.g., <0.5%) with acceptable confidence intervals. A discussion on whether this number provides a stable proportion of homozygous and heterozygous informative SNPs in their target population is warranted.

5. Low-Level Detection and Accuracy

In post-transplant monitoring—particularly after lung transplantation—dd-cfDNA levels are typically below 1% in the absence of rejection. As shown in Table 1, the measured values at mixing rates of 0.5% and 1% deviate noticeably from the actual values. I recommend increasing the density of sample testing within the 0%–1% mixing range to validate the method's reliability for low-level dd-cfDNA quantification.

6. Use of Pre/Post-Transplant Difference Metric

The use of the differential allele ratio (post-transplant minus pre-transplant) for clustering is a central part of the method when baseline data are available. The authors assume this subtracts out background noise. However, this approach is highly sensitive to batch effects and technical variations between sequencing runs. The manuscript would be strengthened by providing quality control data demonstrating the technical reproducibility of the platform between runs, or by employing and validating an alternative normalization strategy.

7. Substitute for Pre-Transplant Samples

The manuscript states: "If a pre-transplant recipient sample is not available, leukocyte-derived genomic DNA or DNA from other sources can be fragmentase-treated and used as a substitute." However, cfDNA and leukocyte-derived gDNA differ significantly in fragment length, structure, and distribution. How do the authors address the potential impact of these differences on the accuracy and reliability of the results?

8. Sample Size and Statistical Power

Parent-Child Pairs: Given the limited sample size of only three parent-child pairs (n=3), conclusions drawn from this analysis should be treated with caution, and this limitation should be clearly stated in the Discussion.

Table 2: Lacks explicit indication of sample sizes, which should be added for clarity and transparency.

9. Comparison with Existing Methods

Although other methods are mentioned in the Introduction, it would be best to more directly compare this method with the gold standard in this field (such as ArcherDX’s LiquidIQ or Natera’s Prospera), highlighting its advantages and potential disadvantages in terms of process simplification, cost, and speed.

Moreover, the authors present their clustering method as a novel solution for dd-cfDNA quantification without donor genotype. However, the study fails to benchmark its performance against established donor-independent algorithms, such as:

Maximum likelihood estimation-based method using a binomial model [2]. A direct comparison on the same dataset(s) is essential to demonstrate the relative advantages, disadvantages, and potential improvements offered by the clustering approach in terms of accuracy, precision, and robustness, especially at low dd-cfDNA fractions.

10. Graphical Enhancements

It is recommended that trend lines or confidence intervals be added to the scatter plots in Figures 2, 3, and 4 to make the correlations more intuitive.

The time series diagram in Figure 5 suggests connecting the results of different methods with points of different shapes to more clearly observe the differences and consistencies among the three methods (direct method, clustering method 1, and clustering method 2). Additionally, Figure 5 lacks a clear legend to distinguish the different data series—adding one would significantly improve readability.

Reference

1. Cao C, Yuan L, Wang Y, Liu H, Cuello Garcia H, Huang H, et al. Analysis of the primary factors influencing donor derived cell-free DNA testing in kidney transplantation. Front Immunol. 2024;15: 1435578. doi:10.3389/fimmu.2024.1435578

2. Zhou Y, Yang G, Liu H, Chen Y, Li X, Ge J, et al. A Noninvasive and Donor-independent Method Simultaneously Monitors Rejection and Infection in Patients With Organ Transplant. Transplant Proc. 2019;51: 1699–1705. doi:10.1016/j.transproceed.2019.04.051

**Do you want your identity to be public for this peer review?** For information about this choice, including consent withdrawal, please see our Privacy Policy

Reviewer #1: **Yes:** Frederik Banch Clausen

Reviewer #2: No

---

## [Author Response · Author response to Decision Letter 1]

12 Oct 2025

Dear Reviewers, We would like to thank the reviewers and editors for their thoughtful and constructive comments, which have helped us significantly improve the clarity, rigor, and clinical relevance of the manuscript. We have carefully addressed each of the reviewers’ comments in a detailed response letter, and have revised the manuscript accordingly.

---

## [Decision Letter · Decision Letter 1]

11 Nov 2025

Dear Dr. Yamada,

Thank you for submitting your manuscript to PLOS ONE. After careful consideration, we feel that it has merit but does not fully meet PLOS ONE’s publication criteria as it currently stands. Therefore, we invite you to submit a revised version of the manuscript that addresses the points raised during the review process.

**The authors have done significant improvement of this submission, however, they need to highlight the cost effectiveness of this method as compared to existing methods, including addressing the minor comments from the second reviewer.**

We look forward to receiving your revised manuscript.

Kind regards,

Elingarami Sauli, PhD

Academic Editor

PLOS ONE

Journal Requirements:

Reviewers' comments:

Reviewer's Responses to Questions

**Comments to the Author**

Reviewer #1: All comments have been addressed

Reviewer #2: (No Response)

2. Is the manuscript technically sound, and do the data support the conclusions?

Reviewer #1: Yes

Reviewer #2: Partly

3. Has the statistical analysis been performed appropriately and rigorously?

Reviewer #1: Yes

Reviewer #2: Yes

4. Have the authors made all data underlying the findings in their manuscript fully available?

Reviewer #1: Yes

Reviewer #2: Yes

5. Is the manuscript presented in an intelligible fashion and written in standard English?

Reviewer #1: Yes

Reviewer #2: Yes

Reviewer #1: I find that the manuscript is ready for accept. The authors have provided a comprehensive and thorough response to the concerns and suggestions raised by the reviewers and satisfactorily amended the manuscript accordingly. I have no further comments.

Reviewer #2: 1. Clinical Correlation and Utility

I appreciate that the authors added a clinical case (S4 Fig) illustrating dd-cfDNA changes. However, in this figure, the orange line (w/o pre-data) lacks data points where dd-cfDNA levels exceed 10%, especially around the event biopsy time point. Since rejection is a dynamic process, missing data at these clinically relevant points limits interpretation. The authors are encouraged to supplement these data if available.

In addition, the figure currently lacks axis labels for both x- and y-axes. Each figure should be self-explanatory so that readers can understand its meaning even without referring to the caption.

Finally, while I understand that a direct comparison with existing dd-cfDNA assays was beyond the current scope, such a comparison would be highly valuable for future work, as it would strengthen the clinical relevance and translational potential of this new method.

2. Cost and Efficiency

I would like to update the authors’ understanding regarding existing methods: commercially available dd-cfDNA assays that do not require donor genotyping are already in widespread use.

In the Discussion, the authors briefly mention relative cost and efficiency advantages, but I could not find detailed information about cost aspects in the revised manuscript. Since cost and reliability are key determinants for clinical adoption, a short discussion on the economic feasibility or expected cost differences compared to existing commercial assays would be appreciated.

3. Population Specificity and Generalizability

Thank you for your clarification.

4. SNP Panel Design and Sequencing Parameters

Thank you for your clarification.

5. Low-Level Detection and Accuracy

While I understand the authors’ rationale, the potential measurement error in the 0–1% dd-cfDNA range remains clinically concerning. For example, in lung transplantation, a 1% threshold is often used to indicate possible rejection. In Table 1, when the actual mixing rate is 0.5%, the measured value is 1.0%, which could lead to significant clinical misinterpretation. This limitation should be clearly discussed, as small inaccuracies around this threshold can greatly influence clinical decision-making.

6–8. Use of Pre/Post-Transplant Difference Metric, Substitute for Pre-Transplant Samples, Sample Size and Statistical Power

Thank you for your responses.

9. Comparison with Existing Methods

Our earlier comment was intended to encourage comparison with existing commercial dd-cfDNA platforms to highlight the advantages and limitations of the new method. Although the authors indicated that direct comparison was difficult, I still suggest that, if feasible in future work, they test a subset of samples using a commercial platform to compare turnaround time and measurement concordance. This would significantly strengthen the paper’s translational impact.

10. Graphical Enhancements

Thank you for your response.

**Do you want your identity to be public for this peer review?** For information about this choice, including consent withdrawal, please see our Privacy Policy

Reviewer #1: **Yes:** Frederik Banch Clausen

Reviewer #2: No

---

## [Author Response · Author response to Decision Letter 2]

16 Nov 2025

We appreciate the opportunity to revise our manuscript and thank the Academic Editor and reviewers for their valuable feedback.

We thank the Academic Editor for the constructive guidance. In response to the request to highlight the cost-effectiveness of our method, we have revised the Discussion to provide a clearer explanation of the potential economic advantages of our workflow compared with existing commercial dd-cfDNA assays. Specifically, we now clarify that our approach eliminates donor genotyping and genome-wide mapping steps, relies solely on targeted SNP capture and simple clustering analysis, and can be implemented on standard benchtop NGS platforms without the need for proprietary software. While we refrain from providing numerical estimates because a formal cost analysis was not conducted, we emphasize that these features are expected to reduce reagent consumption, analytical overhead, and operational burden.

In addition, all minor comments raised by Reviewer 2 have been fully addressed. In particular, we added a statement noting that several donor-independent dd-cfDNA assays (e.g., AlloSure, Prospera) are already commercially available and widely used, and we revised the text to more clearly describe how our workflow relates to these existing approaches.

In accordance with the Reviewer’s and the Editor’s request for improved visualization of the correlations, we have added three new supplementary figures (S1–S3) showing regression lines and 95% confidence intervals for all correlation analyses originally presented in Figure 4(a–c). These supplementary figures enhance the clarity of the linear relationships and directly address the request for graphical improvements. No further modifications to the main figures were necessary.

We believe these revisions satisfactorily address the Editor’s and Reviewers’ comments and improve the overall clarity, context, and presentation of our method.

---

## [Editor Report · Decision Letter 2]

15 Dec 2025

Dear Dr. Yamada,

Thank you for submitting your manuscript to PLOS ONE. After careful consideration, we feel that it has merit but does not fully meet PLOS ONE’s publication criteria as it currently stands. Therefore, we invite you to submit a revised version of the manuscript that addresses the points raised during the review process.

Please address reviewer 2's comments from the previous decision letter.

We look forward to receiving your revised manuscript.

Kind regards,

Elingarami Sauli, PhD

Academic Editor

PLOS One
---

## [Author Response · Author response to Decision Letter 3]

22 Dec 2025

We sincerely thank Reviewer #2 for the careful re-evaluation of our revised manuscript and for the constructive and insightful comments. We believe that addressing these points has substantially improved the clarity, clinical relevance, and translational positioning of our work. Our detailed responses are provided below.

1. Clinical Correlation and Utility

Reviewer comment:

The orange line (w/o pre-data) in Supplementary Figure S4 lacks data points exceeding 10%, particularly around the biopsy-proven rejection event. In addition, axis labels are missing. A comparison with existing assays would also strengthen clinical relevance.

Response:

We thank the reviewer for these important observations.

First, we have revised Supplementary Figure S4 to include explicit axis labels for both axes (x-axis: Time relative to transplantation (days); y-axis: dd-cfDNA (%)), ensuring that the figure is fully self-explanatory without reference to the caption.

Regarding the absence of >10% dd-cfDNA data points in the w/o pre-data (orange) curve, we would like to clarify that these values were not missing from the dataset but were intentionally not displayed in the plotted time series due to reduced estimation reliability in this range. Specifically, when pre-transplant reference data are unavailable, higher dd-cfDNA fractions may reduce cluster separability in the recipient-only clustering framework, leading to less robust estimation. To avoid over-interpretation of potentially unstable values, these estimates were omitted from the plotted curve and instead reported in the figure legend for transparency.

To maintain transparency, we now report the corresponding w/o pre-data estimates in the legend (e.g., 8.55% at day 176 and 10.79% at day 252), while explaining why values in this range are not displayed in the figure. We also emphasize in the Discussion that future prospective studies—including denser longitudinal sampling and, where feasible, cross-platform comparisons with established commercial dd-cfDNA assays—will be important to further validate performance across the full dynamic range.

2. Cost and Efficiency

Reviewer comment:

Donor-independent commercial dd-cfDNA assays are already in widespread use, and the revised manuscript lacks sufficient discussion of cost considerations.

Response:

We appreciate the reviewer’s clarification and apologize for any ambiguity in our original wording. We have revised the manuscript to clearly acknowledge that donor-independent commercial dd-cfDNA assays are already widely available and clinically implemented.

Although we did not perform a formal cost analysis, we have expanded the Discussion to provide a methodological perspective on economic feasibility. Specifically, we now explain that our workflow is based on a targeted SNP capture panel (~300 loci) and simple clustering-based analysis, eliminating donor genotyping and genome-wide sequencing. This design reduces sequencing depth requirements and allows implementation on standard benchtop NGS platforms using open-source software, in contrast to commercial assays that rely on proprietary algorithms and specialized infrastructures.

We emphasize that these considerations suggest potential advantages in operational simplicity and scalability, while refraining from quantitative cost claims. A formal head-to-head cost and efficiency comparison with commercial platforms is identified as an important objective for future work.

3. Population Specificity and Generalizability

Reviewer comment:

Thank you for your clarification.

Response:

We thank the reviewer for the acknowledgment and have no further changes to add.

4. SNP Panel Design and Sequencing Parameters

Reviewer comment:

Thank you for your clarification.

Response:

We appreciate the reviewer’s confirmation and have no additional modifications regarding this point.

5. Low-Level Detection and Accuracy

Reviewer comment:

Measurement error in the 0–1% dd-cfDNA range may lead to clinically significant misinterpretation, particularly around commonly used thresholds such as 1%.

Response:

We thank the reviewer for highlighting this clinically important issue and fully agree that uncertainty in the low dd-cfDNA range warrants careful discussion.

In response, we have added a dedicated paragraph to the Discussion explicitly addressing measurement variability in the 0–1% dd-cfDNA range. As noted by the reviewer, in Table 1 a nominal mixing ratio of 0.5% resulted in an estimated dd-cfDNA value of approximately 1.0%. We clarify that this variability reflects stochastic variation in read sampling and cluster assignment that becomes more pronounced at very low allele fractions.

Importantly, we now emphasize that dd-cfDNA values close to commonly used clinical decision thresholds should not be interpreted in isolation, but rather evaluated in conjunction with longitudinal trends and the overall clinical context. We note that such considerations are particularly relevant in transplant settings where clinical decision thresholds around this range are commonly applied. We further emphasize that the primary strength of the proposed clustering-based framework lies in capturing relative temporal changes within individual patients, rather than providing absolute precision at the lowest detectable range.

Finally, we note that future methodological refinements, increased sequencing depth, and prospective validation studies will be required to further improve robustness and accuracy in this clinically sensitive low-level range.

6–8. Use of Pre/Post-Transplant Difference Metric, Substitute for Pre-Transplant Samples, Sample Size and Statistical Power

Reviewer comment:

Thank you for your responses.

Response:

We appreciate the reviewer’s acknowledgment and have no further changes to add.

9. Comparison with Existing Methods

Reviewer comment:

A future comparison with commercial dd-cfDNA platforms would significantly strengthen translational impact.

Response:

We fully agree with the reviewer. While direct comparison with commercial platforms was beyond the scope of the current study, we have now explicitly addressed this point in the Discussion. Specifically, we acknowledge the widespread clinical use of donor-independent commercial dd-cfDNA assays and clarify that our clustering-based approach is intended as a complementary and transparent workflow rather than a direct replacement. We further state that future prospective studies evaluating a subset of samples using both our method and established commercial assays will be essential to assess analytical concordance, turnaround time, and appropriate clinical use cases.

10. Graphical Enhancements

Reviewer comment:

Thank you for your response.

Response:

We thank the reviewer for the positive feedback and confirm that the requested graphical clarifications have been implemented.

---

## [Editor Report · Decision Letter 3]

19 Jan 2026

A SNP-Based Capture and Clustering Workflow to Assess Donor-Derived Cell-Free DNA in Transplantation

PONE-D-25-42729R3

Dear Dr. Yamada,

We’re pleased to inform you that your manuscript has been judged scientifically suitable for publication and will be formally accepted for publication once it meets all outstanding technical requirements.

Kind regards,

Elingarami Sauli, PhD

Academic Editor

PLOS One
---

## [Editor Report · Acceptance letter]

PONE-D-25-42729R3

PLOS One

Dear Dr. Yamada,

I'm pleased to inform you that your manuscript has been deemed suitable for publication in PLOS One. Congratulations! Your manuscript is now being handed over to our production team.

Kind regards,

on behalf of

Dr. Elingarami Sauli

Academic Editor

PLOS One